# Shotgun Proteomics Revealed Preferential Degradation of Misfolded In Vivo Obligate GroE Substrates by Lon Protease in *Escherichia coli*

**DOI:** 10.3390/molecules27123772

**Published:** 2022-06-11

**Authors:** Tatsuya Niwa, Yuhei Chadani, Hideki Taguchi

**Affiliations:** Cell Biology Center, Institute of Innovative Research, Tokyo Institute of Technology, Yokohama 226-8503, Japan; tniwa@bio.titech.ac.jp (T.N.); chadani.y.aa@m.titech.ac.jp (Y.C.)

**Keywords:** molecular chaperone, chaperonin, GroEL, protease, Lon protease, proteomics, proteostasis

## Abstract

The *Escherichia coli* chaperonin GroEL/ES (GroE) is one of the most extensively studied molecular chaperones. So far, ~80 proteins in *E. coli* are identified as GroE substrates that obligately require GroE for folding in vivo. In GroE-depleted cells, these substrates, when overexpressed, tend to form aggregates, whereas the GroE substrates expressed at low or endogenous levels are degraded, probably due to misfolded states. However, the protease(s) involved in the degradation process has not been identified. We conducted a mass-spectrometry-based proteomics approach to investigate the effects of three ATP-dependent proteases, Lon, ClpXP, and HslUV, on the *E. coli* proteomes under GroE-depleted conditions. A label-free quantitative proteomic method revealed that Lon protease is the dominant protease that degrades the obligate GroE substrates in the GroE-depleted cells. The deletion of DnaK/DnaJ, the other major *E. coli* chaperones, in the ∆*lon* strain did not cause major alterations in the expression or folding of the obligate GroE substrates, supporting the idea that the folding of these substrates is predominantly dependent on GroE.

## 1. Introduction

Most proteins must fold into their native structures to gain their functions [1]. However, protein folding often competes with the formation of aggregates, which not only impair protein function but also cause cytotoxicity in some cases [2,3,4]. In cells, molecular chaperones facilitate the proper folding of various proteins by preventing aggregate formation [2,3]. Indeed, previous studies have revealed that a significant fraction of the proteins in any cell requires at least one or several chaperones. In *Escherichia coli*, GroEL/ES (GroE) and DnaK/DnaJ (DnaKJ)-GrpE are known to assist in the proper folding of a large subset of proteins [2,3].

GroE is the only essential chaperone for the growth of *E. coli* [5,6]. One of the decades-long efforts in chaperonin biology is to identify the substrate proteins that are obligately dependent on GroE for proper folding in the cell (in vivo obligate GroE substrates). Kerner et al. [6] identified hundreds of proteins that interact with GroEL in *E. coli*, using a mass spectrometry (MS)-based proteomics approach. The GroEL-interactors are classified into three categories (Classes I, II, and III) according to the relative amount of the proteins bound to GroEL against the total cellular amount of each protein. In their study, the Class III substrates, which are most enriched in the GroE complex, were classified as potential obligate GroE substrates. Subsequently, Fujiwara et al. [7] conducted a further systematic assessment by using a conditional GroE expression strain, *E. coli* MGM100 [8], and found that only a subset of the Class III substrates has obligate GroE dependences for their folding in vivo. These substrates are regarded as in vivo obligate GroE substrates, and termed Class IV substrates. Further analyses based on data from a cell-free proteomics approach identified several in vivo obligate GroE substrates [9].

Many of these in vivo obligate GroE substrates formed aggregates when overexpressed in the GroE-depleted cells [7]. In addition to the aggregate formation, a subset of the GroE substrates expressed under leaky conditions are degraded in the GroE-depleted cells [7]. This degradation is thought to be a consequence of the failure of the substrates to fold under the GroE-depleted conditions, resulting in targets for proteases. In the *E. coli* cytosol, three ATP-dependent proteases, Lon, ClpXP, and HslUV, are involved in the degradation of misfolded proteins [10,11,12]. However, the physiological differences of these proteases in the clearance of misfolded proteins derived from chaperone deficiencies at the proteome level are largely unknown.

To evaluate the abundance of a wide range of proteins in cells, comprehensive and deep quantitative proteomic analysis technologies are needed. Over the past two decades, shotgun proteomics using liquid chromatography coupled to tandem MS (LC-MS/MS) has achieved remarkable progress, including precise quantitative analyses such as SILAC [13]. SILAC requires stable isotopic labeling, which restricts the available media for cultivation. Recent advances in label-free quantification technologies, such as a method based on the signal intensity of the MS1 ion chromatogram (so-called LFQ intensity) [14] and DIA/SWATH [15], have expanded the options for medium selection and culture conditions. Using the SWATH-MS acquisition method, we aim to investigate the “fate” of the in vivo obligate GroE substrates under GroE-depleted conditions. Specifically, the degradation properties of these substrates when misfolded was evaluated by deletions of three major cytosolic proteases (Figure 1A). The results showed that most of the obligate GroE substrates were degraded by Lon protease in the GroE-depleted cells. Further analysis using a DnaKJ-deleted strain and its derivative again revealed that GroE is predominantly involved in the folding of most obligate GroE substrates.

## 2. Results

### 2.1. Obligate GroE Substrates Tend to Be Degraded by Lon under GroE-Depleted Conditions

To investigate the degradation properties of the in vivo obligate GroE substrates under GroE-depleted conditions, we employed the SWATH-MS acquisition method. SWATH-MS can evaluate the relative abundances of proteins in a label-free manner at the proteome level in nutrient-rich media. To deplete the expression of GroE in cells, we used the MGM100 strain, in which the *groESL* gene is controlled by the arabinose-inducible *BAD* promoter to regulate the expression by sugar, as used in previous studies [7,9] (Figure 1A). First, we compared the expression amounts of the whole-cell proteins under GroE-depleted conditions with those under GroE-normal conditions. Based on this analysis, we evaluated the fold changes of ~1300 proteins (Appendix A). Among these evaluated proteins, ~20 in vivo obligate GroE substrates were included (Table 1). As reported previously, the MS analysis confirmed the GroE depletion-induced large proteome alteration, including the over expression of MetE and several heat shock proteins such as ClpB and DnaK [7] (Appendix A). Importantly, the expression of the obligate GroE substrates showed a strong tendency to decrease when GroE was depleted (Figure 1B). This result suggests that many obligate GroE substrates tend to be degraded, rather than forming aggregates in the GroE-depleted cells.

Next, we investigated the relevance of the Lon, ClpXP, and HslUV cytosolic proteases on the degradation of misfolded proteins in the GroE-depleted cells. We made new deletion strains of each protease in MGM100 as the background strain (MGM100Δ*lon*, MGM100Δ*clpPX*, and MGM100Δ*hslVU*). Then, we investigated their proteome changes elicited by the GroE-depletion. Strikingly, the volcano plot, which indicates the variation of expression amounts on the horizontal axis and statistical certainty on the vertical axis, showed that the decrease in the obligate GroE substrates under GroE-depleted conditions was largely recovered in the MGM100Δ*lon* strain (Figure 1C and Appendix A). This recovery trend was statistically significant from the fold change distributions of the obligate GroE substrates (*p* = 0.000942, by Wilcoxson’s rank-sum test), represented as a boxplot, which depicted the distribution of the values in each sample (Figure 1D). In addition, some of the previously known Lon substrates including LipA, one of the obligate GroE substrates, were increased in the MGM100Δ*lon* strain (Appendix A), corroborating the assumption based on the previous findings. In contrast, the volcano plots of the ClpXP and HslUV deletion strains did not show this trend (Figure 1C and Appendix A). The boxplot revealed a weak recovery of the GroE substrates in the MGM100Δ*hslVU* cells, but the difference was not statistically significant (*p* = 0.3273, by Wilcoxson’s rank-sum test) (Figure 1D). No significant changes by GroE depletion were observed for the GroE substrates belonging to other classes (Class I, II, and III minus) in any of the *E. coli* strains (Appendix A). Furthermore, the MS analysis revealed that five GroE substrates (FbaB, FtsE, NagZ, YbhA, and Tas), which were identified in the GroE-normal cells but not in the GroE-depleted cells were reproducibly identified in the MGM100Δ*lon* strain under GroE-depleted conditions (Table 1 and Appendix A). This result suggests that the proteolysis of the five proteins by Lon protease under GroE-depleted conditions was circumvented in the Lon-deleted cells.

### 2.2. Deletion of DnaKJ Barely Affects the Folding of Most In Vivo Obligate GroE Substrates

Our previous reconstituted cell-free translation (PURE system) analysis revealed that almost all of the obligate GroE substrates have a strong tendency to form aggregates without chaperones [7,16], but for many of them the aggregation-formation is rescued by the DnaKJ system [17]. Therefore, we assumed that the folding of the in vivo obligate GroE substrates might depend on not only GroE but also DnaKJ. If this is the case, then the Lon deletion in DnaKJ-deleted cells would affect the abundance of the in vivo obligate GroE substrates, even in the presence of GroE. Accordingly, we compared the proteome changes between the wildtype, *dnaKJ*-deleted (Δ*dnaKJ*), and *dnaKJ*&*lon*-deleted (Δ*dnaKJ*Δ*lon*) strains. As reported previously, the deletion of *dnaKJ* caused drastic proteome changes [18] (Figure 2A and Appendix A). Among ~1300 evaluated proteins, 60~70 and 60~80 proteins were specifically up- and down-regulated (fold change > 2 or <0.5 and adjusted *p*-value < 0.05) by the deletion of *dnaKJ* or *dnaKJ* and *lon*, respectively (Figure 2A and Appendix A). In contrast, the proteome change by the deletion of *lon* in addition to *dnaKJ* was small (Figure 2A and Appendix A). Although 30~40 proteins were specifically up-regulated (fold change > 2 and adjusted *p*-value < 0.05) by the deletion of *lon*, their fold change values were not large compared to the results in wildtype vs. Δ*dnaKJ* or wildtype vs. Δ*dnaKJ*Δ*lon* (Figure 2A and Appendix A). In addition, only about five to nine proteins were down-regulated in the Δ*dnaKJ*Δ*lon* strain against Δ*dnaKJ* (Figure 2A and Appendix A). Notably, the obligate GroE substrates did not show any remarkable changes in both deletion strains (Figure 2A, Table 2 and Appendix A). This result suggests that DnaKJ is not an additional factor associated with the folding of the obligate GroE substrates in cells.

However, the possibility that these substrates form aggregates before degradation remained, and hence the amounts of the proteins are not changed. To assess this possibility, we prepared the pellet fraction from the lysates of each strain by centrifugation and conducted the same proteomic analysis. Although the reproducibility of the fold change values was not as good as that of the total proteome analysis (Appendix A), the results clearly demonstrated that only a small subset of the obligate GroE substrates accumulated in the Δ*dnaKJ* cells and the Δ*dnaKJ*Δ*lon* cells (Figure 2B and Table 2). In other words, the absence of DnaKJ does not induce the aggregate formation of many obligate GroE substrates. In summary, the results suggest that there is no additional benefit to having DnaKJ for the folding of many obligate GroE substrates when GroE is present.

### 2.3. Metabolic Perturbations by Protease Deletions under GroE-Depleted Conditions Revealed by Clustering Analysis

In the above analyses, we only focused on the changes in the amounts of the obligate GroE substrates. However, although the GroE depletion alone causes drastic changes in protein expression and metabolism [7], the additional deletion of the proteases may elicit further perturbations of the proteome or metabolome. Therefore, we assessed the proteome changes caused by deleting each protease under GroE-depleted conditions. For this purpose, we performed a clustering approach with the fold change values, defined as the ratio of protein abundances under GroE-normal and GroE-depleted conditions in each strain. We chose ~1000 proteins, with fold change values quantified in all four strains, for clustering by the k-means method. The number of clusters was set to six, and the fold change values were converted to logarithmic values before the clustering. The clustering analysis returned four clusters containing small numbers of proteins (Clusters 1~4) and two clusters with larger numbers of proteins (Clusters 5~6) (Figure 3A and Appendix A). Clusters 1~4 exhibited relatively more significant differences in their fold-changes than the other two, suggesting that these four clusters could provide some information about the specific changes caused by the deletion of the proteases. We then applied an enrichment analysis with annotation by the KEGG BRITE hierarchy [19] to characterize these four clusters (Table 3). As shown in Table 3, many metabolic pathways were enriched in each cluster. Especially, Cluster 3, in which the fold change pattern in MGM100Δlon was only decreased, had the highest number of metabolic pathways with fluctuations, including amino acid metabolism and nucleic acid metabolism. This result suggests that the deletion of Lon protease in the GroE-depleted cells causes a further metabolic perturbation in addition to the GroE depletion.

To investigate the perturbations from the deletion of Lon more directly, we defined the proteins with specific changes between MGM100 and MGM100Δ*lon* from the distribution of the fold changes (Figure 3B). The results of the enrichment analysis for the up-regulated and down-regulated proteins in MGM100Δ*lon* showed that the deletion of Lon caused the up-regulation of some metabolic enzymes related to amino acid synthesis under GroE-depleted conditions (Table 4 and Appendix A).

Another minor change was observed in Cluster 1, as its fold change pattern revealed a large increase in both MGM100Δ*lon* and MGM100Δ*hslVU* as compared to the other two strains. This cluster included some proteins induced in the stationary growth phase, such as Sra, Dps, WrbA, ElaB, and OsmC.

## 3. Discussion

In this analysis, we have shown that many in vivo obligate GroE substrates are degraded by cytosolic proteases under GroE-depleted conditions, and Lon protease is mainly responsible for this degradation (Figure 1). Conversely, DnaKJ does not act as a dominant factor for the folding of these substrates, although a few obligate GroE substrates tended to form aggregates by the deletion of DnaKJ (Figure 2). Based on these results, a plausible scheme of the behavior of the obligate GroE substrates under GroE-depleted conditions is depicted in Figure 4. When GroE is absent, these substrates cannot complete their folding and are degraded by proteases such as Lon. In contrast, when DnaKJ was absent, most of the obligate GroE substrates can complete their folding with the aid of GroE, although a small fraction of the obligate GroE substrates form aggregates in cells. However, since various chaperones such as GroE are highly up-regulated in Δ*dnaKJ* cells [18], our observation might be affected by these up-regulated chaperones’ effects. Considering this point, our results do not exclude the possibility that DnaKJ is also involved in the folding of the obligate GroE substrates under some conditions.

The statistical analysis of the proteome changes in our experiments suggested that the deletion of Lon may cause additional metabolic perturbations, such as in amino acid synthesis. Note that the *lon* deletion does not show large proteome changes under nutrient-rich medium conditions (Niwa T. et al., in preparation). Of course, since the GroE depletion itself reportedly induces large metabolic changes, including the depletion of several amino acids, S-adenosylmethionine, and NADPH [7], this phenomenon may be significant only under exceptional metabolic conditions. However, this observation might reflect the possibility that protein degradation affects the metabolism in an unappreciated manner, although it may be significant only under extreme conditions such as severe energy deficiency.

In addition, HslUV may also be involved in the degradation of misfolded proteins and additional metabolic perturbations, as shown in Figure 1B,C and Figure 3A and Table 3. Although HslUV is expressed abundantly in cells and its expression is induced by heat stress, knowledge about the physiological role of HslUV is limited. Our observations suggest that HslUV may have certain specific functions in the cell, with some overlapping with Lon. However, the evidence for this overlapped role is not strong, and the details are still unclear.

The up-regulation of some stationary-phase-induced proteins in both MGM100Δ*lon* and MGM100Δ*hslVU* suggests that the deletion of these two proteases may cause additional changes related to starvation or another specific factor could invoke the transition of the growth phase under GroE-depleted conditions. However, the connection between them remains unclear. We confirmed that RpoS, a factor responsible for various stress responses and the growth phase transition [20,21], does not appear to be involved in this change since its expression was up-regulated only in MGM100Δ*clpPX* (Appendix A). Accordingly, further investigations are needed to clarify the physiological roles and the overlapped manners of these cytosolic proteases in detail.

In summary, our study has partially uncovered the fate of the in vivo obligate GroE substrates under GroE-depleted conditions. Furthermore, the inability to degrade the misfolded protein perturbs proper intracellular metabolism. The intimate link between chaperones and proteases in cellular proteostasis is important but not well understood; hence, the approach conducted here would be valuable for analyses of other organisms, including eukaryotes.

## 4. Materials and Methods

### 4.1. Bacterial Strains

*E. coli* strains used in this study are listed in Appendix A. The DNA fragment amplified from JW0013-KC (∆*dnaK*::FRT-Km^R^-FRT) [22], using the primers PT0071 (AAATTGGGCAGTTGAAACCAGAC) and PM0195 (GATGTTTCGCTTGGTGGTCGAATGGGCAGG), and that from JW0014-KC (∆*dnaJ*::FRT-Km^R^-FRT), using the primers PT0072 (TACAGGTGCTCGCATATCTTCAACG) and PM0196 (CCTGCCCATTCGACCACCAAGCGAAACATC), were mutually annealed and amplified using PT0071 and PT0072. The purified DNA was electroporated into the *E. coli* strain BW25113 harboring pKD46 [23], and the transformants resistant to 40 µg/mL kanamycin were stored as ECY0262 (BW25113∆*dnaKJ*, Km^R^).

The DNA fragment amplified from JW3903-KC (∆*hslV*::FRT-Km^R^-FRT), using the primers PT0223 (CCATCTATAATTGCATTATG) and PM0195, and that from JW3902-KC (∆*hslU*::FRT-Km^R^-FRT), using the primers PT0224 (CTGAGTTCGGCTAATTTGTTG) and PM0196, were mutually annealed and amplified using PT0223 and PT0224. The purified DNA was electroporated into the *E. coli* strain BW25113 harboring pKD46, and the transformants resistant to 40 µg/mL kanamycin were stored as ECY0289 (BW25113∆hslVU, Km^R^).

The DNA fragment amplified from JW0427-KC (∆*clpP*::FRT-Km^R^-FRT), using the primers PT0065 (ATGGTGATGCCGTACCCATAACAC) and PM0195, and that from JW0428-KC (∆*clpX*::FRT-Km^R^-FRT), using the primers PT0066 (AGCCCGATCCGCCATCTAACTTAGC) and PM0196, were mutually annealed and amplified using PT0065 and PT0066. The purified DNA was electroporated into the *E. coli* strain BW25113 harboring pKD46, and the transformants resistant to 40 µg/mL kanamycin were stored as ECY0290 (BW25113∆clpPX, Km^R^).

Amplified pKD3 [23] plasmid DNA was electroporated into ECY0289, ECY0290, and JW0429 (∆*lon*::FRT-Km^R^-FRT) harboring pCP20A (modification of pCP20 [23], with an inactivated cat selection marker), and the transformants resistant to 20 µg/mL chloramphenicol were stored as ECY0292 (∆*hslVU*, Cm^R^), ECY0293 (∆*clpPX*, Cm^R^), and ECY0210 (∆*lon*, Cm^R^), respectively.

Phage P1-mediated transduction was used to introduce the ∆*dnaKJ*, ∆*hslVU*, ∆c*lpPX* and ∆*lon* mutations from ECY0262, ECY0292, ECY0293 and ECY0210, respectively.

### 4.2. Cell Culture and Sample Preparation for the LC-MS/MS Analysis

For the analysis of the MGM100 and MGM100-derived mutants, cells were grown in LB medium supplemented with 0.2% arabinose at 37 °C and harvested at an early logarithmic growth phase (0.2~0.3 OD_660_). After washing with LB medium, the cells were inoculated into LB medium supplemented with 1 mM diaminopimelate and either 0.2% arabinose or 0.2% glucose. At this time, the OD_660_ of the culture solution was set to 0.04. After 3 h of cultivation at 37 °C, the cells were harvested. The OD_660_ at the time of collection was around 0.8~1.0. For the analyses of Δ*dnaKJ* and Δ*dnaKJ*Δ*lon*, cells were grown in LB medium at 37 °C and harvested at a logarithmic growth phase (~1.0 OD_660_).

The harvested cells were resuspended in the PTS solution [24] (100 mM Tris-HCl (pH 9.0), 12 mM sodium deoxycholate, 12 mM sodium N-lauroylsarcosinate) and boiled at 95 °C for 5 min. The solution was then frozen at −80 °C for 10 min. Next, the solution was sonicated in an ultrasonic bath for 20 min at room temperature for further cell disruption. After the cell disruption, the protein concentration was measured using the BCA protein assay kit (Thermo Fisher Scientific, Waltham, MA, USA) and fixed at 50 μg in 100 μL or 25 μg in 50 μL by dilution with the PTS solution.

The obtained total proteins were reduced by 10 mM dithiothreitol and incubated for 30 min at room temperature. Afterwards, 50 mM iodoacetamide was added, and the solution was incubated for 20 min at room temperature in the dark for alkylation. After the reduction and alkylation, the solution was diluted 5-fold with 50 mM ammonium bicarbonate and digested by adding Lys-C protease (FUJIFILM-Wako, Osaka, Japan) at 1/100 of the total protein weight and incubating for 3 h at room temperature. The fragmented peptides were further digested by adding Trypsin Gold (Promega, Madison, Wisconsin, USA) at 1/50 of the total protein weight, and incubated at 37 °C overnight. After the digestion, an equal volume of ethyl acetate and 1/20 volume of 10% trifluoroacetic acid (TFA) were added to the peptide solution and mixed vigorously. The mixture was centrifuged at 15,700× *g* for 2 min, and the upper ethyl acetate layer containing the surfactants was withdrawn. The resulting lower water layer was dried with a centrifugal evaporator. The peptides were then re-dissolved in 0.1% TFA and 2% acetonitrile and desalted with a handmade Stage Tip [25] composed of an SDB-XC Empore Disk (3 M, Maplewood, MN, USA). The peptides bound to the Stage Tip were eluted with 0.1% TFA and 80% acetonitrile. After the desalting, the peptides were dried by a centrifugal evaporator again and re-dissolved in 0.1% TFA and 2% acetonitrile for LC-MS/MS measurements.

For the proteome analysis of pellet fractions, cells were resuspended in lysis buffer (50 mM Tris-HCl (pH 7.5), 100 mM NaCl, 1 mM EDTA) supplemented with a protease inhibitor cocktail (cOmplete™ mini, EDTA free, Roche, Basel, Switzerland) and disrupted by sonication. The resulting lysate was centrifuged at 20,000× *g* for 10 min, and the supernatant was discarded. After washing twice with the lysis buffer, the pellet was dissolved in the PTS solution. The protein concentration was measured with a BCA protein assay kit and fixed at 10 μg in 25 μL by dilution with the PTS solution. The subsequent processes were performed as described above.

### 4.3. LC-MS/MS Measurement and Data Analysis

The LC-MS/MS measurements were conducted with an Eksigent NanoLC Ultra and TripleTOF 4600 tandem-mass spectrometer or an Eksigent NanoLC 415 and TripleTOF 6600 mass spectrometer (AB Sciex, Framingham, MA, USA). The trap column used for nanoLC was a 5.0 mm × 0.3 mm ODS column with a particle size of 5 μm (L-column2, Chemical Evaluation and Research Institute, Tokyo, Japan). The separation column was a 12.5 cm × 75 μm capillary column packed with 3 μm C18-silica particles (Nikkyo Technos, Tokyo, Japan). The detailed settings for the LC-MS/MS measurements are summarized in Appendix A. The measurement was conducted three times for each sample. One biological replicate set was used for the analysis of the MGM100 and its derivative strains, and two biological replicate sets were used for Δ*dnaKJ* and Δ*dnaKJ*Δ*lon* strains.

Data analysis was performed by the DIA-NN software (version 1.7.16, https://github.com/vdemichev/diann, accessed on 30 April 2021) [26]. The library for SWATH-MS was obtained from the SWATH atlas (http://www.swathatlas.org/, accessed on 30 April 2021); the original data were acquired by Midha et al. [27]. The fold changes between mean intensities and *p*-values by Welch’s *t*-test were calculated by an in-house R script (R.app for Mac, version 3.6.2). For the correction of multiple testing, *p*-values were adjusted by the Benjamini-Hochberg method (using the “p.adjust” function). Only the proteins with intensities obtained in all three measurements in both samples were used to calculate fold changes. The enrichment analysis was performed with an in-house R script (using Fisher’s exact test). The KEGG BRITE hierarchy information (*E. coli* MG1655 strain) was downloaded from the website (https://www.genome.jp/brite/eco00001) on 23 March 2016.

## Figures and Tables

**Figure 1 molecules-27-03772-f001:**
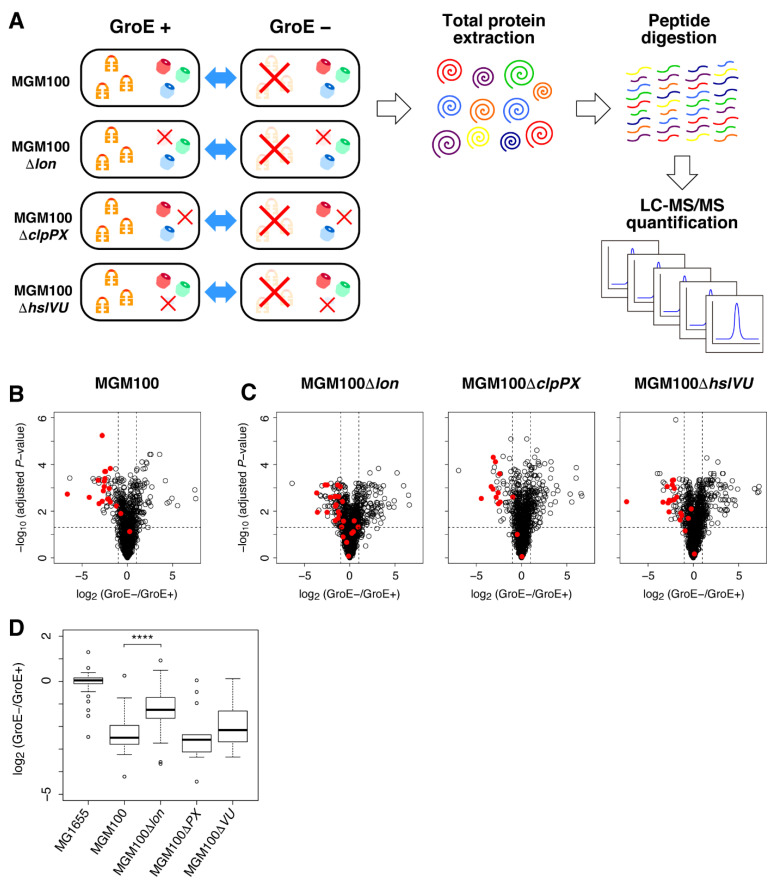
Proteome changes of the in vivo obligate GroE substrates by GroE depletion. (**A**) Schematic illustration of the experiment. The wildtype cells and the protease deletion variants were grown under the GroE-normal and GroE-depleted conditions. The total proteins were then extracted and digested into peptides by the Lys-C/Trypsin proteases. The digested peptides were measured and quantified by LC-MS/MS. (**B**) Proteomic changes in MGM100 by GroE depletion represented as a volcano plot. The horizontal axis indicates the value of the fold changes by the GroE depletion taken as log_2_, and the vertical axis indicates *p*-values by Welch’s *t*-test (two-sided) with three technical replicates in each sample taken as −log_10_. For multiplicity correction, the *p*-values are adjusted by the Benjamini-Hochberg method. Red dots indicate the in vivo obligate GroE substrates. (**C**) Proteomic changes in the MGM100-derived protease-deletion strains by the GroE depletion, depicted as volcano plots. Red dots indicate the in vivo obligate GroE substrates. (**D**) Distribution of the fold changes of the in vivo obligate GroE substrates in each strain depicted as a boxplot. The box portions and the central bands are described according to the 25th percentile and the median, respectively. **** *p*-value < 0.001 (Wilcoxon’s rank-sum test).

**Figure 2 molecules-27-03772-f002:**
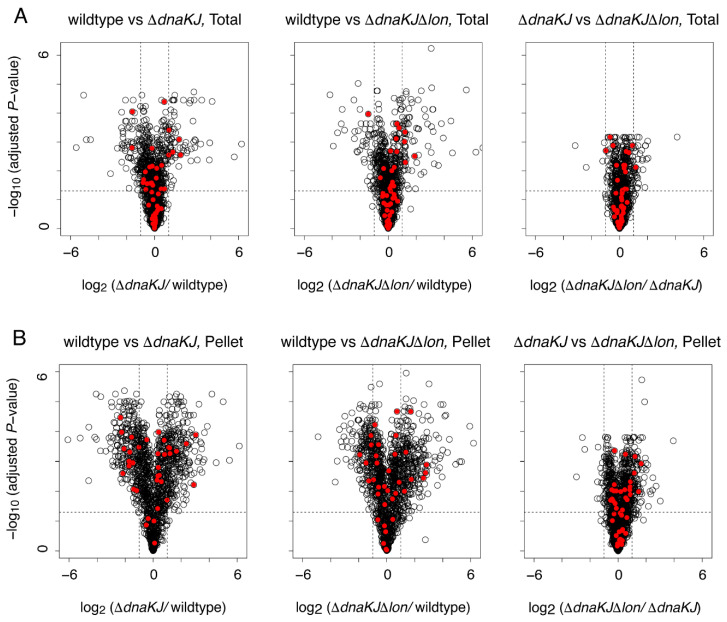
Proteome changes of the in vivo obligate GroE substrates by DnaK/DnaJ deletion. (**A**) Proteome changes of Δ*dnaKJ* and Δ*dnaK*JΔ*lon* in the total fraction depicted as volcano plots. The horizontal axis indicates the value of the fold changes taken as log_2_, and the vertical axis indicates *p*-values by Welch’s *t*-test (two-sided) with three technical replicates in each sample taken as −log_10_. For multiplicity correction, the *p*-values are adjusted by the Benjamini-Hochberg method. Red dots indicate the in vivo obligate GroE substrates. (**B**) Proteome changes of Δ*dnaKJ* and Δ*dnaKJ*Δ*lon* in the pellet fraction depicted as volcano plots. Red dots indicate the in vivo obligate GroE substrates.

**Figure 3 molecules-27-03772-f003:**
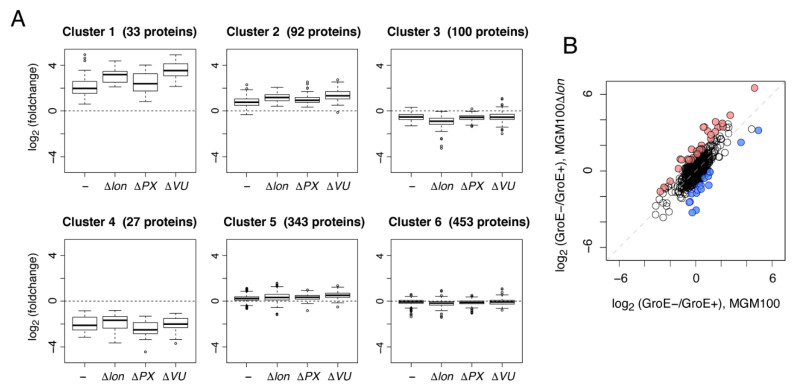
Clustering analysis to investigate the proteome perturbations induced by the protease deletions. (**A**) Distribution of the fold changes of the MGM100 and MGM100-derived protease-deletion strains in each cluster. Fold changes were defined as the protein abundance ratios under GroE-normal and GroE-depleted conditions in each strain. The box portions and the central bands are described according to the 25th percentile and the median, respectively. (**B**) Fold change comparison between MGM100 and MGM100Δ*lon*. Red dots indicate the proteins specifically up-regulated by the Lon deletion, and blue dots indicate the proteins specifically down-regulated by the Lon deletion.

**Figure 4 molecules-27-03772-f004:**
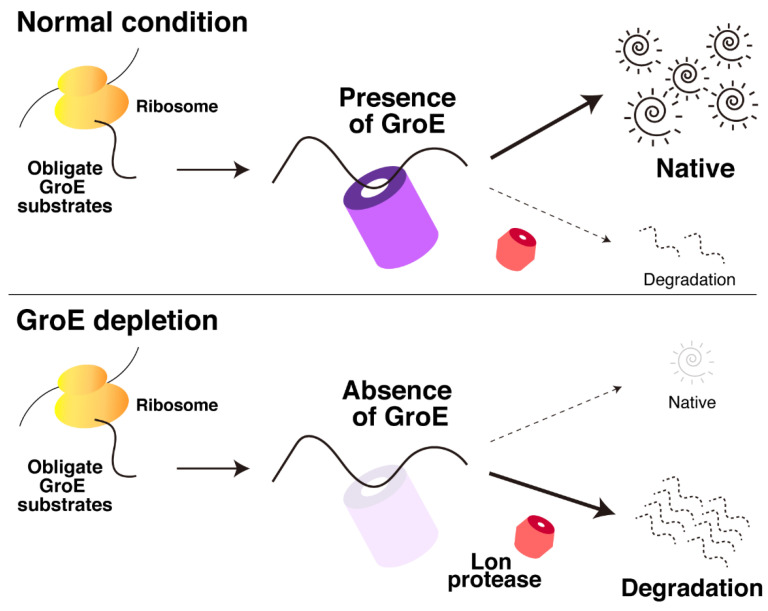
Schematic illustration of the fate of in vivo obligate GroE substrates in GroE-depleted cells. Under normal conditions, the in vivo obligate GroE substrates can fold into their native structures with the aid of GroE. Upon GroE depletion, these GroE substrates tend to be degraded by proteases, preferentially by the Lon protease, since they cannot complete their folding without GroE.

**Table 1 molecules-27-03772-t001:** Fold change values of the in vivo obligate GroE substrates in MGM100 and MGM100-derived protease-deletion strains.

Gene Name	Kerner 2005	Fujiwara 2010	Niwa 2016	MGM100			MGM100Δ*lon*			MGM100Δ*clpPX*			MGM100Δ*hslVU*		
				Number of Detection		FC	Number of Detection		FC	Number of Detection		FC	Number of Detection		FC
				Ara	Glc		Ara	Glc		Ara	Glc		Ara	Glc	
*argP*	3	4		3	3	1.191	3	3	1.407	3	3	1.033	3	3	1.089
*ltaE*	3	4		3	3	0.604	3	3	1.170	3	3	0.723	3	3	0.857
*metK*	3	4		3	3	0.433	3	3	1.902	3	3	0.513	3	3	0.683
*add*	3	4		3	3	0.271	3	3	0.316	3	3	0.194	3	3	0.280
*dapA*	3	4		3	3	0.269	3	3	0.483	3	3	0.187	3	3	0.234
*asd*	3	4		3	3	0.249	3	3	0.569	3	3	0.171	3	3	0.419
*rfbC*	3	4		3	3	0.235	3	3	0.450	3	1		3	3	0.386
*serC*	2	4		3	3	0.221	3	3	0.428	3	3	0.167	3	3	0.363
*hemB*	3	4		3	3	0.188	3	3	0.395	3	3	0.112	3	3	0.202
*pmbA*	3	4		3	3	0.178	3	3	0.225	3	0		3	3	0.176
*lipA*	3	4		3	3	0.177	3	3	0.410	3	3	0.136	3	3	0.201
*nuoC*	2	4		3	3	0.170	3	3	0.080	3	3	0.115	3	3	0.221
*pepQ*	3	4		3	3	0.169	3	3	0.177	3	3	0.143	3	3	0.155
*fabF*	3	4		3	3	0.158	3	3	0.336	3	2		3	3	0.228
*kdsA*	2	4		3	3	0.145	3	3	0.328	3	3	0.098	3	3	0.157
*pyrD*		4		3	3	0.145	3	3	0.176	3	2		3	2	
*gatY*	3	4		3	3	0.114	3	3	0.150	3	3	0.046	3	3	0.098
*deoA*	3	4		3	3	0.106	3	1		3	0		3	3	0.145
*sdhA*	3	4		3	3	0.054	3	3	0.084	3	0		3	3	0.135
*araA*	3	4		3	3	0.010	3	1		3	2		3	3	0.006
*fbaB*	3	4		3	0		3	3	1.323	3	1		3	0	
*ftsE*	3	4		3	0		3	3	0.620	3	0		3	2	
*nagZ*	3	4		3	0		3	3	0.539	3	0		3	0	
*ybhA*			°	3	0		3	3	0.332	3	1		3	0	
*tas*	2		°	3	0		3	3	0.788	3	0		3	0	
*nagD*			°	3	2		3	3	0.598	3	0		3	2	
*alaA*	3	4		3	1		3	3	0.418	3	0		3	3	0.535
*dadA*	3	4		3	1		3	3	0.350	3	0		3	0	

**Table 2 molecules-27-03772-t002:** Fold change values of the in vivo obligate GroE substrates for Δ*dnaKJ* and Δ*dnaKJ*Δ*lon* strain.

Gene Name	Kerner 2005	Fujiwara 2010	Niwa 2016	Number of Detection			Fold Change			Number of Detection			Fold Change		
				wt Total	Δ*KJ* Total	Δ*KJ* Δ*lon* Total	Δ*KJ/*wt Total	Δ*KJlon*/wt Total	Δ*KJlon*/Δ*KJ* Total	wt ppt	Δ*KJ* ppt	Δ*KJ* Δ*lon* ppt	Δ*KJ*/wt ppt	Δ*KJlon*/wt ppt	Δ*KJlon*/Δ*KJ* ppt
*dadX*	3	4		3	3	3	3.611	3.691	1.022	3	3	3	5.136	7.144	1.391
*nanA*	3	4		3	3	3	3.351	1.711	0.511	3	3	3	1.047	0.904	0.864
*sdhA*	3	4		3	3	3	2.466	2.230	0.905	3	3	3	1.725	1.578	0.915
*lldD*	3	4		3	3	3	2.035	1.477	0.726	3	3	3	1.218	0.890	0.731
*dadA*	3	4		3	3	3	2.022	2.320	1.147	3	3	3	1.754	1.847	1.053
*gdhA*		4		3	3	3	1.632	2.291	1.404	3	3	3	2.147	3.361	1.566
*metK*	3	4		3	3	3	1.612	1.008	0.625	3	3	3	1.264	0.996	0.788
*nuoC*	2	4		3	3	3	1.423	1.232	0.866	3	3	3	1.311	1.102	0.840
*ybjS*	3	4		3	3	3	1.395	1.592	1.141	3	3	3	2.299	2.393	1.041
*rfbC*	3	4		3	3	3	1.378	1.079	0.783	3	3	3	8.265	6.135	0.742
*yqaB*	3	4		3	3	3	1.209	1.323	1.094	2	3	3			0.820
*ycfH*	3	4		3	3	3	1.206	0.935	0.775	3	0	2			
*lsrF*	3	4		3	3	3	1.154	1.381	1.196	3	3	3	0.713	0.855	1.200
*ybhA*			°	3	3	3	1.135	1.197	1.055	3	3	3	7.462	6.802	0.912
*gatY*	3	4		3	3	3	1.114	1.541	1.384	3	3	3	2.262	3.266	1.443
*fabF*	3	4		3	3	3	1.088	0.910	0.836	3	3	3	0.496	0.592	1.194
*nagD*			°	3	3	3	1.077	1.073	0.996	3	3	3	1.293	0.892	0.690
*deoA*	3	4		3	3	3	1.047	0.931	0.888	3	3	3	0.443	0.644	1.454
*nagZ*	3	4		3	3	3	1.045	0.954	0.914	3	0	2			
*add*	3	4		3	3	3	1.039	1.135	1.092	3	3	3	3.164	2.571	0.813
*ftsE*	3	4		3	3	3	1.038	1.127	1.086	3	3	3	1.270	1.051	0.827
*lipA*	3	4		3	3	3	1.032	1.343	1.302	3	3	3	1.360	1.327	0.976
*kdsA*	2	4		3	3	3	1.004	1.006	1.002	3	3	3	0.303	0.555	1.829
*alaA*	3	4		3	3	3	0.989	1.054	1.066	3	3	3	1.987	1.499	0.754
*asd*	3	4		3	3	3	0.962	1.086	1.130	3	3	3	0.368	0.666	1.809
*pmbA*	3	4		3	3	3	0.943	1.051	1.114	3	3	3	1.500	1.662	1.109
*serC*	2	4		3	3	3	0.927	1.160	1.251	3	3	3	0.223	0.409	1.837
*argP*	3	4		3	3	3	0.920	1.339	1.456	3	3	3	1.252	1.548	1.236
*pepQ*	3	4		3	3	3	0.916	1.127	1.230	3	3	3	0.199	0.458	2.297
*ltaE*	3	4		3	3	3	0.890	1.192	1.339	3	3	3	0.344	0.615	1.787
*pyrD*		4		3	3	3	0.886	0.838	0.947	3	3	3	0.728	0.667	0.916
*dapA*	3	4		3	3	3	0.879	0.949	1.080	3	3	3	0.210	0.466	2.217
*uxaC*	3	4		3	3	3	0.870	0.772	0.887	3	2	3		0.263	
*tldD*	3	4		3	3	3	0.859	0.944	1.100	3	3	3	0.784	0.966	1.232
*hemB*	3	4		3	3	3	0.812	1.531	1.887	3	3	3	0.303	0.949	3.135
*pyrC*	2	4		3	3	3	0.785	0.797	1.015	3	2	3		0.520	
*argE*	3	4		3	3	2	0.745			3	3	3	1.076	1.346	1.251
*nfo*	3	4		3	3	3	0.686	0.676	0.984	3	0	2			
*yajO*	3	4		3	3	3	0.648	0.802	1.239	3	0	0			
*tas*	2		°	3	3	3	0.633	1.002	1.583	3	1	0			
*gpr*	2		°	3	3	3	0.581	0.756	1.300	3	3	3	0.236	0.651	2.762
*cysH*			°	3	3	3	0.566	1.257	2.221	3	3	2	0.380		
*fbaB*	3	4		3	3	3	0.332	0.372	1.120	3	0	0			
*frdA*	3	4		3	3	3	0.326	0.373	1.147	3	3	3	0.315	0.359	1.141
*yafD*	3	4		0	3	3			1.206	1	3	3			1.461
*fadA*		4		2	3	3			0.740	1	3	3			1.068
*yigB*			°	2	2	2				2	3	3			1.542
*yjhH*		4								0	3	3			1.219
*dusB*	3	4								1	3	3			1.119
*eutB*	3	4		1	0	1				1	3	3			0.962

**Table 3 molecules-27-03772-t003:** Enrichment analysis with the KEGG BRITE hierarchy for each cluster.

Cluster	Annotation (KEGG BRITE Hierarchy3)	Odds Ratio	*p*-Value *	Number of Proteins in Population	Number of Proteins in Subgroup
Cluster 1	Selenocompound metabolism	24.95	0.0009	7	3
	Monobactam biosynthesis	16.15	0.0133	6	2
	Folate biosynthesis	9.23	0.0302	9	2
	Cysteine and methionine metabolism	8.83	0.0008	25	5
	Lysine biosynthesis	8.08	0.0370	10	2
	Glycine, serine and threonine metabolism	5.93	0.0086	27	4
Cluster 2	Histidine metabolism	31.93	0.0025	4	3
	Nitrogen metabolism	10.55	0.0407	4	2
	Chaperones and folding catalysts	5.41	0.0002	31	10
	Starch and sucrose metabolism	4.77	0.0215	13	4
	Glutathione metabolism	3.90	0.0357	15	4
Cluster 3	C5-Branched dibasic acid metabolism	Inf	0.0090	2	2
	Lysine degradation	Inf	0.0090	2	2
	Photosynthesis proteins	16.48	0.0003	8	5
	Oxidative phosphorylation	11.54	5.5 × 10^−7^	21	11
	Tetracycline biosynthesis	9.61	0.0476	4	2
	Citrate cycle (TCA cycle)	6.64	0.0016	15	6
	Purine metabolism	5.33	1.2 × 10^−5^	42	14
	Pyrimidine metabolism	4.89	0.0003	31	10
	One carbon pool by folate	4.84	0.0463	9	3
	Propanoate metabolism	3.90	0.0369	14	4
	Enzymes	1.96	0.0015	561	68
Cluster 4	Bacterial chemotaxis	Inf	0.0006	2	2
	Galactose metabolism	43.52	1.8 × 10^−8^	15	7
	Monobactam biosynthesis	20.12	0.0090	6	2
	Lysine biosynthesis	10.07	0.0253	10	2
	Glycine, serine and threonine metabolism	5.18	0.0295	27	3
	Enzymes	3.12	0.0076	561	21

* *p*-value was calculated by Fisher’s exact test. Only the annotations whose *p*-values were less than 0.05 are listed.

**Table 4 molecules-27-03772-t004:** Enrichment analysis for the proteins with specific changes by the deletion of Lon protease.

Group	Annotation (KEGG BRITE Hierarchy3)	Odds Ratio	*p*-Value *	Number of Proteins in Population	Number of Proteins in Subgroup
Up-regulated in MGM100Δ*lon*	Bacterial motility proteins	Inf	0.0344	1	1
	Monobactam biosynthesis	14.69	0.0158	6	2
	Selenocompound metabolism	11.76	0.0216	7	2
	Cysteine and methionine metabolism	7.96	0.0012	25	5
	Lysine biosynthesis	7.35	0.0434	10	2
	Glyoxylate and dicarboxylate metabolism	6.02	0.0213	18	3
	Glycine, serine and threonine metabolism	5.36	0.0117	27	4
Down-regulated in MGM100Δ*lon*	Glycosylphosphatidylinositol (GPI)-anchored proteins	Inf	0.0162	1	1
	Aminobenzoate degradation	Inf	0.0162	1	1
	Glyoxylate and dicarboxylate metabolism	14.38	0.0025	18	3

* *p*-value was calculated by Fisher’s exact test. Only the annotations whose *p*-values were less than 0.05 are listed.

## Data Availability

The mass spectrometry proteomics data have been deposited in the jPOST repository [38] (https://repository.jpostdb.org/, accessed on 30 April 2021), with reference number JPST001558.

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
