# Peer review of "Shotgun Proteomics Revealed Preferential Degradation of Misfolded In Vivo Obligate GroE Substrates by Lon Protease in Escherichia coli"

_molecules, 2022, doi:10.3390/molecules27123772_

Round 1

Reviewer 1 Report

The manuscript by Niwa et al, Misfolded in vivo obligate GroE substrates are preferentially degraded by Lon protease in Escherichia coli uses a new and interesting approach to examine the fate of GroE substrates.  The first examine the proteomic effects of GroE shutdown, then additionally examine the impact of deletion of each of the three bacterial protease systems.  The data is interesting, but the manuscript would be improved by better presentation of the data. 

  1. The discussion of obligate substrates in the introduction and how they were classified a such is helpful, but a table within the manuscript that clearly lists the substrates that are shown in Fig. 1 is needed, preferably one that also provides details on expression changes in the different backgrounds. These are currently just shown as unlabled red dots.
  2. The discussion of the proteomic data would benefit from an introductory description that summarizes the range of expression changes in the ~1300 proteins examined and clearly states the number of GroE substrates (obligate or otherwise) that were in the dataset. As it is, the supplementary data provides more information on the proteins that were NOT in the dataset then those that were in the dataset (Supp. Table 1).
  3. There are two cases where the authors say something but no citation is provided: " As reported previously, the deletion of dnaKJ caused drastic proteome changes (Figure 2A and Supplementary Dataset S2) " and As reported previously, the MS analysis confirmed the GroE depletion-induced large proteome alteration, including the overexpressions of MetE and several heat shock proteins such as ClpB and DnaK (Supplementary Dataset  S1).  Moreover, if results from these datasets are already published, it would be helpful to clearly distinguish what is different between the current study and the prior study.
  4. Also, since the authors are vague about the identity of the GroE subtrates that are examined, it is not clear whether the same substrates are red dots in both Figs. 1 and 2.
  5. Since the authors did not find a role for DnaJ/K for the obligate GroE substrates it would be interesting if they did see a role of the other classes of GroE substrates that were discussed in the introduction.

Author Response

see attached pdf.

Reviewer 2 Report

GroESL is a highly conserved chaperonin that prevents protein aggregation and promotes protein folding. GroE is the only essential chaperone for growth in E.Coli. GroE has four classes of substrates, and a subset of Class IV substrates are named obligate substrates. The low or endogenous levels of these substrates are degraded in GroE-depleted strains. In this study, the authors used a quantitative proteomics (SWATH-MS) approach to test which of the three major AAA (Lon, ClpXP, HslUV) proteases are responsible for the degradation of GroE obligate substrates. The authors use knock-out strains of these proteases in GroE-depleted conditions using the arabinose inducible GroE strain, MGM100. The authors show that when GroE is depleted, the GroE-obligate substrate levels decrease, suggesting that they cannot fold and are subsequently degraded. However, when they looked at GroE-depletion in ∆lon cells, the GroE-obligate substrate levels did not change significantly. Thus the authors concluded that Lon must be the main protease that degrades GroE-obligate substrates. 

Additionally, the authors compared the proteomic profile of  âˆ†dnaKJ and ∆lon∆dnaKJ cells to wild-type cells and showed the removal of Lon on top of DnaKJ did not change the proteome profile to a high degree. The authors also claimed that since the GroE-obligate substrate does not change in ∆dnaKJ cells compared to wild-type cells, DnaKJ must not be significantly involved in refolding GroE-obligate substrates. However, they did not control GroE levels in this experiment, and in addition, previous studies showed that in ∆dnaKJ strains, GroE is up-regulated. As a result, their observation might be simply that GroE itself was able to refold its substrates in the absence of DnaKJ and the discussion of these results should take this into account. 

Major Comments

  • The methods need to be updated with the number of biological replicates for each condition used.  For example, the currently written methods state how many technical (MS runs) replicates of each sample were taken, but not how many biological replicates were used.
  • For Figure 1, there is clearly a shift of some GroE obligate substrates in the ∆lon strain, however, others were not.  The supplemental datasets were not provided to this reviewer, but it would be important in the main text to point out which obligate GroE substrates seemed to be Lon substrates and which did not. 
  • The authors use ∆dnaKJ and ∆lon∆dnaKJ cells with wild-type GroE and suggest that since the GroE substrates do not show a remarkable difference in ∆dnaKJ strain compared to GroE-depleted cells, then the DnaKJ must not be involved in their refolding in a significant degree even though their previous paper showed that DnaKJ rescues aggregation of GroE substrates in vitro. This in vivo interpretation based solely on the proteomics data is quite a leap, given the presence of wild-type GroE in ∆dnaKJ strain.  Similarly, in the discussion section, the authors acknowledge that several chaperones are up-regulated, including GroE, in ∆dnaKJ strains based on other studies that may be confounding the interpretation. Given this, the authors should soften claims from the abstract and Section 2.2 saying that DnaKJ is not important for folding of GroE substrates, rather specifically describe their results as relaying that there is no additional benefit to having DnaKJ when native levels of GroE are present, or similar words to that effect (for example, line 142 and others).
  • This study is a computational analysis based on proteomics with no in vivo or in vitro follow-up. Considering this fact, the language of the title and paper should be adjusted accordingly to emphasize that this work is hypothesis-generating, rather than conclusive in and of itself. 

Minor Comments:

  • Authors suggest 5 GroE substrates (FbaB, FtsE, NagZ, YbhA, and Tas) are identified in MGM100∆lon cells but not MGM100 cells under GroE-depleted conditions. Based on not finding these proteins in MGM100 cells, they conclude that these were being degraded by Lon under GroE depleted conditions. Ideally these claims should be verified using Western blots or some other means.
  • The authors discuss proteome changes in ∆dnaKJ and ∆lon∆dnaKJ and suggest that changes are small when Lon is deleted together with DnaKJ. The authors should add quantification and grouping to show the scale of differences. (Ex. Number of proteins that change in each group, etc.)
  • Section 2.3. The authors use fold changes obtained from protease-depleted strain in GroE-depleted backgrounds. For clarity, the authors should state the comparison strain used to obtain the fold changes. 
  • Line 170-189.  Can any of these changes be attributed to known Lon substrates (such as SulA, RcsA, etc?).  On a similar note, are any of the known GroE obligate substrates (shown in Figure 1) already known to be Lon substrates? 

Author Response

see attached pdf.

Reviewer 3 Report

The article is written in a precise manner and can be accepted for publication. English needs to be addressed; in some places, long sentences have been used that need to be restructured.

1.      Most proteins must fold into their native structures to gain their functions . However, protein folding often competes with the formation of aggregates, which not only  impair protein function but also cause cytotoxicity in some cases In cells, molecular chaperones facilitate the proper folding of various proteins by preventing aggregate formation. Introduction s section can be improved citing latest references and also adding up a paragraph about the disorders wherein aggregation plays a key role.

2.      A paragraph about the objective of the study needs to be added at the end of introduction section.

3.      In this analysis, we have shown that many in vivo obligate GroE substrates are degraded by cytosolic proteases. Check for the italicising in vivo and in vitro throughout the manuscript.

4.      I found erratic usage of abbreviations throughout the manuscript that are difficult to follow. The authors are advised to make consistent usage of abbreviations throughout the manuscript. 

Author Response

see attached pdf.
